Original research

# Psychological correlates of free colorectal cancer screening uptake in a Scottish sample: a cross-sectional observational study

Chloe Fawns-Ritchie [ORCID],[1] Christopher B Miller,[2] Marjon van der Pol,[3] Elaine Douglas [ORCID],[4] David Bell,[4] Ronan E O'Carroll,[5] Ian J Deary[1]

REO and IJD contributed equally.

[1]Department of Psychology, The University of Edinburgh, Edinburgh, UK
[2]Department of Psychology, University of Stirling, Stirling, UK
[3]Health Economics Research Unit, University of Aberdeen, Aberdeen, UK
[4]Division of Economics, University of Stirling, Stirling, UK
[5]Division of Psychology, University of Stirling, Stirling, UK

**Correspondence to**
Dr Chloe Fawns-Ritchie;
c.fawns-ritchie@ed.ac.uk

## ABSTRACT

**Objectives** Colorectal cancer (CRC) screening uptake in Scotland is 56%. This study examined whether psychological factors were associated with CRC screening uptake.

**Design** Cross-sectional observational study.

**Setting** This study used data from the Healthy AGeing In Scotland (HAGIS) pilot study, a study designed to be representative of Scottish adults aged 50 years and older.

**Participants** 908 (505 female) Scottish adults aged 50–80 years (mean age=65.85, SD=8.23), who took part in the HAGIS study (2016–2017).

**Primary and secondary outcome measures** Self-reported participation in CRC screening was the outcome measure. Logistic regression was used to test whether scores on measures of health literacy, cognitive ability, risk aversion, time preference (eg, present oriented or future oriented) and personality were associated with CRC screening when these psychological factors were entered individually and simultaneously in the same model.

**Results** Controlling for age, age-squared, sex, living arrangement, and sex*living arrangement, a one-point increase in risk aversion (OR=0.66, 95% CI 0.51 to 0.85) and present orientation (OR=0.86, 95% CI 0.80 to 0.94) was associated with reduced odds of screening. Higher scores on health literacy (OR per one-point increase=1.20, 95% CI 1.09 to 1.31), cognitive ability (OR per SD increase=1.51, 95% CI 1.25 to 1.81) and the intellect personality trait (OR per one-point increase=1.05, 95% CI 1.01 to 1.09) were associated with increased odds of screening. Higher risk aversion was the only psychological variable that was associated with CRC screening participation when all psychological variables were entered in the same model and remained associated with CRC screening when additionally adjusting for deprivation and education. A backward elimination model retained two psychological variables as correlates of CRC screening: risk aversion and cognitive ability.

**Conclusion** Individuals who are more risk averse are less likely to participate in free, home CRC screening.

## INTRODUCTION

Colorectal cancer (CRC) is the third most common cancer in Scotland.[1] Screening to detect CRC using the faecal occult blood

### Strengths and limitations of this study

► This study used data from the Healthy AGeing In Scotland pilot study, which is a study designed to be representative of Scottish adults aged 50 years and older.

► A wide range of psychological characteristics were measured that allowed us to investigate whether each psychological characteristic was associated with colorectal cancer screening independent of the other psychological characteristics measured.

► The outcome measure used was whether participants have ever completed a colorectal screening test, and we did not examine whether participants regularly take part in colorectal cancer screening.

test (FOBT) has been found to reduce CRC-related mortality.[2] The Scottish Bowel Screening Programme was implemented in 2009 and involves inviting adults aged 50–74 years to take part in free home screening (FOBT kits are mailed to eligible individuals and returned postage free) for faecal blood every 2 years.[3] Uptake of CRC screening in Scotland is 56%.[3 4] Understanding the characteristics of individuals who do not complete the FOBT are important for designing methods for improving uptake. Younger age, male sex and lower socioeconomic status are associated with lower screening rates.[4 5]

Psychological variables may also influence screening uptake. Many studies have used the Health Belief Model[6] to investigate whether perceptions about the susceptibility and severity of cancer and the perceived benefits and barriers to screening influence cancer screening uptake. The Health Belief Model purports that an individual will participate in a health behaviour, such as cancer screening, if they believe: (1) they are *susceptible* to the condition; (2) the condition is *severe*;

(3) that there would be *benefits* to engaging in a certain health behaviour; and (4) that any benefits to the behaviour outweigh any *barriers*.[6–8] A study of 335 adults aged 50–79 years in the USA tested whether constructs of the Health Belief Model were associated with FOBT as well as other recommended CRC screening procedures in the USA.[9] Specific barriers to completing the FOBT were associated with lower odds of having had a FOBT in the last 12 months; however, perceived CRC risk, perceived benefits of CRC screening in general and perceived benefits of the FOBT were not associated with having had a FOBT in the last 12 months. The Health Belief Model has also been used to predict participation in other types of cancer screening. A review of studies using the Health Belief Model to predict uptake in mammography and pap screening found strong support that perceived benefits were positively associated with screening uptake and perceived barriers were negatively associated with screening uptake; however, they found weak support for an association between perceived susceptibility and severity with screening uptake.[10]

Whereas a large number of studies have tested whether the constructs of the Health Belief Model are associated with cancer screening, the relationship between other psychological characteristics, such as cognitive ability and personality, with cancer screening is less well understood. Individuals with cognitive impairment are less likely to participate in CRC screening.[11 12] There is some evidence that people who score higher on the personality traits extraversion and conscientiousness have higher rates of cancer screening.[11–15] The role of cognitive ability and personality in CRC screening were investigated together using a nationally representative sample of older adults from England.[15] Better cognitive ability and higher conscientiousness, when measured in the same model, were associated with higher rates of CRC screening participation after adjusting for age and sex. These associations were attenuated and non-significant when additionally adjusting for health literacy and household wealth.[15] Adequate health literacy—the ability to obtain and understand basic health information[16]—has been identified as another possible correlate of CRC screening. Whereas some studies have found that lower health literacy was associated with reduced rates of CRC screening, others have not.[17–20] Gale *et al*[15] tested whether the association between health literacy and CRC screening may be partly explained by cognitive ability. The association between health literacy and CRC screening was attenuated by 40% when adjusting for cognitive ability.[15]

Other psychological characteristics that may be associated with cancer screening include time and risk preference.[21–25] Individuals who are more future oriented may be more likely to participate in routine cancer screening because these individuals place a higher value on future benefits of screening, such as reducing the need for more invasive cancer treatment, and a lower value on the immediate costs of screening, such as time and

effort, compared with individuals who are more present oriented.[22 23] Future oriented women have been found to be more likely to undergo breast cancer and cervical screening.[22–24] However, when examining the role of time preference in prostate exam attendance, more present oriented men were more likely to have had this procedure.[22] Characteristics associated with one screening programme may not necessarily be those associated with other types of screening.[26]

A study of 809 UK adults examined the role of time perspective in the association between socioeconomic inequalities and CRC screening uptake (flexible sigmoidoscopy).[25] This study did not find direct associations between high consideration of future consequences (future orientation) and CRC screening uptake; however, the association between high socioeconomic status and CRC screening uptake was mediated by future orientation. This study also found that future orientation was positively associated with perceived benefits and negatively associated with perceived barriers of CRC screening, which were in turn associated with screening uptake.[25]

More risk averse individuals may be more likely to take part in screening because it can reduce the risk of ill-health by identifying cancer earlier, and therefore, the treatment required is less aggressive and the prognosis is better. Alternatively, risk averse individuals may be less likely to take part in screening because they may see screening as risky.[23 24] Positive screening results are likely to lead to medical treatments that have uncertain outcomes. Studies that have investigated the association between risk aversion and cancer screening provide some support for the latter hypothesis and that risk averse individuals are less likely to participate in cancer screening. More risk averse women have been found to be less likely to undergo regular breast cancer screening.[23] Another study[24] concluded that risk aversion was weakly associated with reduced use of cancer screening services; however, these associations did not reach statistical significance.

Most studies have tended to examine the association between only one psychological variable, or a small number of psychological variables, with cancer screening uptake. These psychological characteristics, however, have moderate correlations with each other. Higher cognitive ability is moderately correlated with health literacy[27] and with greater future orientation.[28] Risk aversion correlates with lower cognitive function and lower literacy.[29] Risk aversion and time preference also have small, but inconsistent, associations with conscientiousness, agreeableness and extraversion.[30] It is not clear whether risk aversion, present orientation, health literacy, cognitive ability and personality traits, which are moderately correlated with each other, have associations with CRC screening uptake that are independent of the other psychological characteristics. Using data from the Healthy AGeing in Scotland (HAGIS) pilot study—a study designed to be

representative of middle-aged and older Scottish adults—the present study aimed to test whether risk aversion, present orientation, cognitive ability, health literacy and personality traits, when examined simultaneously in the same model, have independent associations with self-reported participation in home CRC screening.

## METHODS
### Participants
Data from the HAGIS pilot study[31] were used here. To recruit participants, a random sample of 5211 residential addresses in mainland Scotland were identified using the Postcode Address File. These addresses were screened by the National Records for Scotland to find addresses where at least one resident was aged over 50 years. A total of 3088 addresses were identified with someone aged 50 years and older living in the household, and these were the list of addresses fieldworkers used to recruit HAGIS participants. More information on the sampling procedure is reported elsewhere.[31] Eligible participants for the HAGIS study were those aged over 50 years and their partners if their partner was aged over 45 years.[31] A total of 1057 participants took part in the HAGIS study in 2016–2017 and form the HAGIS sample (figure 1).

Face-to-face interviews were carried out in the participants' own home using Computer Aided Personal Interviewing (CAPI). Written informed consent was obtained prior to the start of the interview. Information collected during the face-to-face interview included health, cognitive function and social circumstances. At the end of the face-to-face interview, participants were also given a self-completion questionnaire that they could complete online, on paper and return via pre-paid post or via the CAPI immediately after the face-to-face interview. The self-completion questionnaire covered topics including health behaviours and personality. Of the 1057 participants who took part in HAGIS, 705 completed the self-completion questionnaire. Data collected during the face-to-face interview and the self-completion questionnaire were used in the current analysis.

### Patient and public involvement
HAGIS participants were not involved in the development of any part of this study. The results of this study will be disseminated to participants and the public via the HAGIS website (www.hagis.scot).

### Measures
#### CRC screening
As part of the face-to-face interview, participants were asked 'Have you ever completed a home testing kit for screening bowel cancer?'. Responses were recorded as 'yes', 'no', 'don't know' or 'refused' to answer.

#### Risk aversion
Risk aversion and time preference were assessed using multiple price lists (MPLs), a standard method for eliciting preferences in economics.[32] An MPL with three choices was used to assess risk aversion in the self-completion questionnaire (shown in online supplemental materials). Participants were presented with a series of choices between a certain amount of monthly income of £1500 or taking a gamble with a 50% chance of a higher monthly income and a 50% chance of a lower monthly income. The lower monthly income increased from £1000 in question 1 to £1300 in question 3. Less risk-averse individuals will be more likely to choose the gamble. The measure of risk aversion was the point at which the participant switched from choosing the certain monthly income of £1500 to choosing the gamble (score range 1–4). Higher scores reflect more risk aversion.

#### Time preference: present orientation
Time preference was assessed using an MPL with seven choices administered in the self-completion questionnaire (shown in online supplemental materials) that were based on a simplified version of the Kirby Delay-Discounting task.[33] Participants had to choose between receiving £1500 now, or a higher amount of money in 1 month's time. The amount received in 1 month's time ranged from £1506 to £1596. If participants are very future oriented, they will always pick the higher amount in 1 month's time. The measure of time preference used here was the point at which the participant switched from choosing the money now to choosing the money in 1 month's time (score range 1–8). A higher score reflects more present orientation.

#### Health literacy
Health literacy was assessed during the face-to-face interview using a popular test of health literacy, the Newest Vital Sign (NVS).[34] The NVS assesses reading and numeracy skills required to understand health information.[34] Participants were presented with a nutrition label from a container of ice cream and were asked six questions about this information. For example, one question asked participants to work out how many calories they would eat if they ate the entire container of ice cream. The score is the sum of correct answers (maximum score=6).

#### Cognitive ability
Five tests of cognitive function that were assessed in the face-to-face interview were used here. These tests were designed to measure verbal declarative memory (immediate and delayed word recall), executive function (categorical fluency), processing speed (letter digit substitution), crystallised ability (vocabulary) and non-verbal reasoning (matrices). Scores on the five cognitive tests were entered into a principal components analysis, and the score on the first unrotated component was saved and used as a measure of general cognitive ability (mean=0.00, SD=1.00). Descriptions of each cognitive test and more detail on the creation of the general cognitive ability measure are reported in the online supplemental file 1.

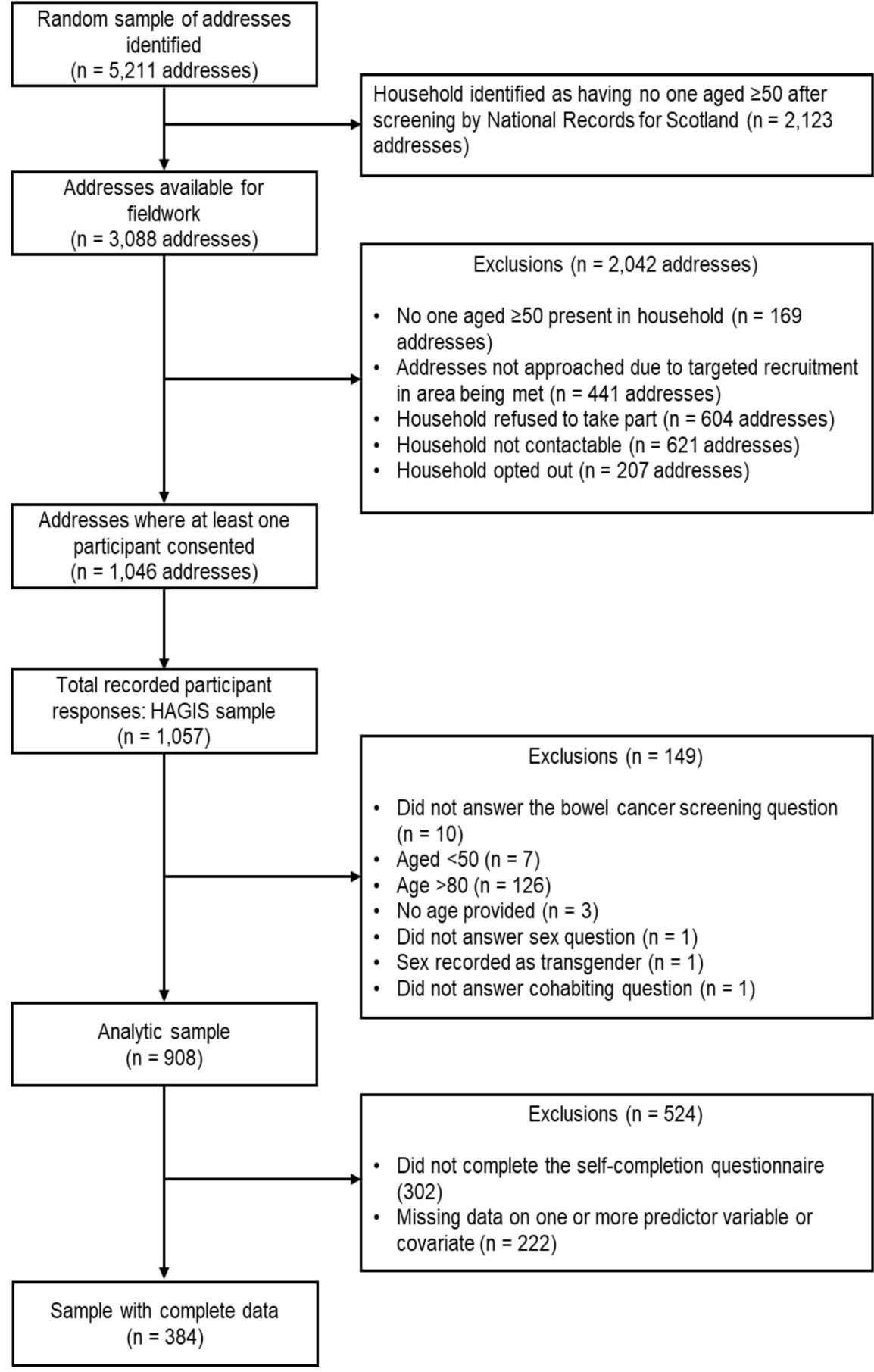

**Figure 1** Participant flow chart. HAGIS, Healthy Ageing In Scotland

## Personality traits

Personality was assessed as part of the self-completion questionnaire using the 50-item International Personality Item Pool (IPIP) questionnaire, which measures the Big Five personality traits of extraversion, agreeableness, conscientiousness, emotional stability (the opposite of neuroticism) and intellect/imagination.[35] This questionnaire consists of 10 items for each personality factor. Participants are presented with a statement (eg, 'I make people feel at ease') and were asked to rate how accurately each statement described them on a 5-point Likert scale (very inaccurate to very accurate). The score for each factor is the sum of these 10 items.

## Covariates

Age in days, sex, living arrangement, social deprivation and educational qualifications were assessed as part of the face-to-face interview. Participants who reported they were married or living with their partner as if married were categorised as cohabiting. Individuals who reported they were single, separated, divorced or widowed were categorised as living alone. Deprivation was measured using Scottish Index of Multiple Deprivation (SIMD) quartiles,[36] which ranks participant's level of deprivation based on the amount of employment, income, health, education, housing, access and crime in the area where they live. Education was assessed by asking participants 'What is the highest level of education you have completed?'. Qualifications were coded as 1=primary or less, 2=O level/O grade or equivalent, 3=highers/sixth year studies/Higher National Certificate (HNC)/Higher National Diploma (HND) or equivalent, 4=first degree and 5=postgraduate/higher degree.

## Statistical analysis

To test whether individuals who have participated in CRC screening differ from non-participants on demographic, socioeconomic and psychological variables, t-tests were performed for normally distributed variables, Mann-Whitney U tests were used for non-normally distributed variables and $\chi^2$ tests were used for categorical variables. Pearson correlations, rank-order correlations and partial correlations adjusting for age and sex were calculated to examine the strength of the relationship between each of the psychological factors of interest. The correlations between covariates was also calculated.

Logistic regression was used to examine the association between psychological variables and CRC screening participation. The outcome variable was coded 1 for reporting participating in CRC screening and 0 if not. Age, age-squared, sex, living arrangement and an interaction between sex and living arrangement were entered as covariates into all models. An age-squared term was added as the assumption of linearity of the logit was violated. An interaction term between sex and living arrangement was included because previous research using the HAGIS data has found a significant interaction between sex and living arrangement on CRC

screening uptake.[37] This study found that single men were less likely to take part in CRC screening, but there were no differences in CRC screening uptake for men living with a partner, single women or women living with a partner.[37]

To test the association of each psychological variable with CRC screening, models were run in which each of the psychological variables (risk aversion, present orientation, health literacy, cognitive ability and each personality trait) were entered individually. Next, deprivation and qualifications were added as covariates to investigate whether the strength of the association between each psychological variable and CRC screening changed after adjustment for indicators of socioeconomic status.

To determine whether any associations between risk aversion, present orientation, health literacy, cognitive ability and personality traits with CRC screening were independent of the other psychological characteristics, a multivariate logistic regression model was run in which all psychological variables were entered into the same model. A fully adjusted model was then run, which additionally adjusted for deprivation and qualifications. A backward elimination logistic regression was run to identify the best independent correlates of CRC screening participation. If the backward elimination model identified the same psychological variables as the models including all psychological variables entered simultaneously, this would provide additional support that this psychological characteristic was an important correlate of CRC screening participation.

Finally, we tested whether there was an interaction between risk aversion and present orientation. Completing a CRC screening test can involve the risk of a bad outcome in the short term, whereas not completing a CRC screening test could involve the risk of a bad outcome in the future. Models were run that included risk aversion, present orientation and the interaction between risk aversion and present orientation, first adjusting for age, age-squared, sex, living arrangement and the interaction between sex and living arrangement and then additionally adjusting for deprivation and qualifications.

For each model, the sample size was based on the number of participants with complete data on all variables entered into the model therefore the sample size for each model varies. Imputation was not used in the current study because those who did and did not return the self-completion questionnaire may differ on the psychological variables assessed in the self-completion questionnaire (eg, conscientiousness). The models were re-run using a subsample of participants with complete data on all variables. The p values reported for the logistic regression models have been false discovery rate (FDR) corrected to control for multiple comparisons. The FDR method controls for the number of false positive results in those tests that reach significance.

## RESULTS

Of the 1057 participants who took part in HAGIS, the following participants were removed before conducting the current analyses: participants who did not answer 'yes' or 'no' to the CRC screening question (no data, n=1; 'don't know', n=6; 'refused', n=3); participants without data on age (n=3), participants aged <50 years (n=7), participants aged >80 years (n=126); participants without data on sex (n=1) or whose sex was recorded as transgender (n=1); and participants without data on cohabiting status (n=1). We removed participants without data on CRC screening, age, sex and living arrangement because these variables were entered into every model. We removed participants aged under 50 years and over 80 years because these participants would not have been invited to take part in the Scottish Bowel Cancer Screening Programme. After removal of these participants, the analytic sample consisted of 908 participants. A flow chart showing how the analytic sample was derived is shown in figure 1.

The mean age of the sample was 65.85 (SD=8.23). A total of 683 (75.2%) participants had completed a home CRC screening test, whereas 225 (24.8%) had not. Participant characteristics according to screening status are shown in table 1. Individuals who participated in CRC screening were older, more likely to be cohabiting, have higher qualifications and were less likely to live in deprived areas, compared with participants who did not participate in CRC screening. Screeners were less risk averse, more future oriented and scored higher on health literacy, cognitive ability, emotional stability and intellect.

Pearson and rank-order correlations between psychological variables and covariates are reported in online supplemental table 1, and partial correlations between the psychological variables controlling for age and sex are reported in online supplemental table 2. Many of the psychological variables have moderate correlations with each other. Notable among them, higher cognitive ability was moderately correlated with higher health literacy (r=0.46, p<0.001; rho=0.45, p<0.001). Risk aversion and present orientation were correlated positively with each other (r=0.14, p=0.002; rho=0.14, p=0.002). Cognitive ability was negatively correlated with risk aversion (r=−0.13, p=0.003; rho=−0.17, p<0.001) and present orientation (r=−0.11, p=0.022; rho=−0.11, p<0.001). Health literacy was negatively correlated with risk aversion (r=−0.12, p=0.004; rho=−0.17, p<0.001). Cognitive ability was positively correlated with all five personality traits (r=0.14 to 0.38, p<0.001; rho=0.17 to 0.39, p<0.001).

Table 2 (model 1) shows the ORs (95% CI) for participating in CRC screening, for each psychological variable entered separately, adjusting for age, age-squared, sex, living arrangement and sex*living arrangement. More risk averse (OR=0.66, 95% CI 0.51 to 0.85) and more present oriented (OR=0.86, 95% CI 0.80 to 0.94) individuals were less likely to participate in CRC screening. Individuals with higher health literacy (OR=1.20, 95% CI 1.09 to 1.31), cognitive ability (OR=1.51, 95% CI 1.25 to 1.81) and intellect (OR=1.05, 95% CI 1.01 to 1.09) scores were more likely to screen. When additionally adjusting for deprivation and qualifications (table 2; model 2), the associations between risk aversion (OR=0.71, 95% CI 0.55 to 0.91), health literacy (OR=1.17, 95% CI 1.06 to 1.28) and cognitive ability (OR=1.36, 95% CI 1.10 to 1.67) were slightly attenuated but remained significantly associated with CRC screening participation. The associations between present orientation (OR=0.91, 95% CI 0.83 to 0.99) and intellect (OR=1.03, 95% CI 0.99 to 1.07) were no longer significant.

Next, all of the psychological variables were entered into the same model, adjusting for age, age-squared, sex, living arrangement and sex*living arrangement (table 3, model 1). The association between risk aversion and CRC screening became slightly stronger (OR=0.52, 95% CI 0.36 to 0.76). Present orientation, health literacy, cognitive ability and the five personality traits were not associated with CRC screening participation when all psychological variables were entered simultaneously. The association between risk aversion and CRC screening remained almost unchanged when additionally adjusting for qualifications and deprivation (OR=0.53, 95% CI 0.36 to 0.77; table 3, model 2). In this fully adjusted model, older age was associated with increased odds of screening. The significant age-squared term suggests that this association became slightly weaker with increased age. Participants living alone, compared with those cohabiting, were less likely to screen. Although the CIs for deprivation and the interaction between sex and living arrangement did not cross 1, these associations were not significant following FDR adjustment.

A backwards elimination logistic regression run from the fully adjusted model retained two psychological variables: risk aversion (OR=0.55, 95% CI 0.39 to 0.78) and cognitive ability (OR=1.42, 95% CI 1.05 to 1.93). This model also retained age, age-squared, living arrangement and SIMD (table 3; model 3).

The interaction between risk aversion and present orientation was non-significant (OR=1.04, 95% CI 0.94 to 1.14; n=494) in a model including risk aversion, present orientation, risk aversion*present-orientation, age, age-squared, sex, living arrangement and sex*living arrangement. This interaction remained non-significant when additionally adjusting for deprivation and qualifications (OR=1.03, 95% CI 0.94 to 1.13; n=490).

Of the 908 participants who make up the analytic sample, 524 individuals had missing data on one or more of the predictor variables and covariates, including 302 participants who did not complete the self-completion questionnaire (figure 1). Demographic characteristics were compared between

**Table 1** Participant characteristics according to colorectal cancer screening participation status

| | N | Yes (n=683) | No (n=225) | Effect size | P value for a difference |
|---|---|---|---|---|---|
| Age (years), mean (SD) | 908 | 66.51 (7.95) | 63.87 (8.76) | r=0.136 | <0.001 |
| Sex, n (%) | 908 | | | Φ=0.011 | 0.741 |
| Female | | 382 (55.9) | 123 (54.7) | | |
| Male | | 301 (44.1) | 102 (45.3) | | |
| Living arrangement, n (%) | 908 | | | Φ=0.122 | <0.001 |
| Cohabiting | | 474 (69.4) | 126 (56.0) | | |
| Living alone | | 209 (30.6) | 99 (44.0) | | |
| Qualifications, n (%) | 902 | | | Φ$_c$=0.123 | 0.009 |
| Primary or less | | 117 (17.3) | 50 (22.3) | | |
| O level/O grade | | 194 (28.6) | 81 (36.2) | | |
| Highers/sixth year studies/HNC/HND | | 208 (30.7) | 53 (23.6) | | |
| First degree | | 90 (13.3) | 29 (12.9) | | |
| Postgraduate/higher degree | | 69 (10.2) | 11 (4.9) | | |
| SIMD, n (%) | 908 | | | Φ=0.163 | <0.001 |
| 1 (Most deprived) | | 139 (20.4) | 73 (32.4) | | |
| 2 | | 136 (19.9) | 59 (26.2) | | |
| 3 | | 213 (31.2) | 50 (22.2) | | |
| 4 (Least deprived) | | 195 (28.6) | 43 (19.1) | | |
| Risk aversion, n (%) | 581 | | | | |
| 1 (Risk seeking) | | 47 (10.3) | 8 (6.5) | | |
| 2 | | 46 (10.1) | 6 (4.8) | | |
| 3 | | 55 (12.0) | 12 (9.7) | | |
| 4 (Risk averse) | | 309 (67.6) | 98 (79.0) | | |
| Risk aversion, mean (SD) | | 3.37 (1.03) | 3.61 (0.85) | r=0.104 | 0.012 |
| Present orientation, n (%) | 504 | | | | |
| 1 (Future oriented) | | 159 (39.7) | 24 (23.3) | | |
| 2 | | 27 (6.7) | 4 (3.9) | | |
| 3 | | 20 (5.0) | 2 (1.9) | | |
| 4 | | 34 (8.5) | 8 (7.8) | | |
| 5 | | 20 (5.0) | 3 (2.9) | | |
| 6 | | 17 (4.2) | 15 (14.6) | | |
| 7 | | 44 (11.0) | 21 (20.4) | | |
| 8 (Present oriented) | | 80 (20.0) | 26 (25.2) | | |
| Present orientation, mean (SD) | | 3.89 (2.89) | 5.15 (2.74) | r=0.159 | <0.001 |
| Health literacy, mean (SD) | 867 | 4.51 (1.68) | 4.07 (1.86) | r=0.103 | 0.002 |
| Cognitive ability, mean (SD) | 790 | 0.06 (0.98) | −0.21 (1.06) | g=0.273 | 0.001 |
| Extraversion, mean (SD) | 575 | 30.64 (7.17) | 30.71 (7.17) | g=0.009 | 0.929 |
| Agreeableness, mean (SD) | 566 | 40.18 (5.95) | 39.88 (6.40) | r=0.010 | 0.804 |
| Conscientiousness, mean (SD) | 547 | 37.77 (6.04) | 36.59 (6.63) | g=0.191 | 0.065 |
| Emotional stability, mean (SD) | 575 | 33.68 (7.22) | 31.99 (7.97) | g=0.229 | 0.035 |
| Intellect, mean (SD) | 565 | 33.53 (5.61) | 32.14 (5.91) | r=0.112 | 0.008 |

g, Hedge's g; HNC, Higher National Certificate; HND, Higher National Diploma; r, effect size correlation coefficient; SIMD, Scottish Index of Multiple Deprivation; Φ, Phi; Φ$_c$, Cramer's V.

participants with and without complete data (online supplemental table 3). Those with complete data were more likely to be cohabiting, had higher qualifications and lived in areas with less deprivation than non-completers. The models reported in table 2 were re-run using participants with complete

**Table 2** ORs (95% CI) for participating in colorectal cancer screening

| | Model 1 | | Model 2 | |
|---|---|---|---|---|
| | n | OR (95% CI) | n | OR (95% CI) |
| Risk aversion | 581 | 0.66 (0.51 to 0.85)** | 577 | 0.71 (0.55 to 0.91)* |
| Present orientation | 504 | 0.86 (0.80 to 0.94)** | 500 | 0.91 (0.83 to 0.99) |
| Health literacy | 867 | 1.20 (1.09 to 1.31)** | 861 | 1.17 (1.06 to 1.28)** |
| Cognitive ability | 790 | 1.51 (1.25 to 1.81)*** | 784 | 1.36 (1.10 to 1.67)* |
| Extraversion | 575 | 0.99 (0.96 to 1.02) | 571 | 0.99 (0.96 to 1.02) |
| Agreeableness | 566 | 1.01 (0.97 to 1.04) | 562 | 1.01 (0.97 to 1.04) |
| Conscientiousness | 547 | 1.03 (1.00 to 1.07) | 543 | 1.03 (0.99 to 1.07) |
| Emotional stability | 575 | 1.03 (1.00 to 1.06) | 571 | 1.02 (0.99 to 1.05) |
| Intellect | 565 | 1.05 (1.01 to 1.09)* | 561 | 1.03 (0.99 to 1.07) |

Model 1 reports the ORs (95% CIs) for each psychological variable entered individually, adjusting for age, age-squared, sex, living arrangement and sex*living arrangement. Model 2 reports the ORs (95% CIs) when additionally adjusting for deprivation and qualifications.
P values are false discovery rate corrected.
*P<0.05, **p<0.01, ***p<0.001.

**Table 3** ORs (95% CIs) for participating in colorectal cancer screening

| | Model 1 (n=388) | Model 2 (n=384) | Model 3 (n=384) |
|---|---|---|---|
| Risk aversion | 0.52 (0.36 to 0.76)** | 0.53 (0.36 to 0.77)** | 0.55 (0.39 to 0.78)** |
| Present orientation | 0.94 (0.85 to 1.04) | 0.98 (0.88 to 1.08) | – |
| Health literacy | 1.02 (0.84 to 1.24) | 1.02 (0.84 to 1.25) | – |
| Cognitive ability | 1.38 (0.97 to 1.97) | 1.26 (0.86 to 1.85) | 1.42 (1.05 to 1.93) |
| Extraversion | 0.97 (0.92 to 1.01) | 0.97 (0.92 to 1.01) | – |
| Agreeableness | 0.95 (0.90 to 1.02) | 0.96 (0.90 to 1.03) | – |
| Conscientiousness | 1.04 (0.99 to 1.10) | 1.04 (0.98 to 1.10) | – |
| Emotional stability | 1.03 (0.98 to 1.07) | 1.02 (0.98 to 1.07) | – |
| Intellect | 1.02 (0.96 to 1.08) | 1.01 (0.95 to 1.08) | – |
| Age | 1.08 (1.04 to 1.12)*** | 1.08 (1.04 to 1.12)** | 1.09 (1.05 to 1.12)*** |
| Age$^2$ | 0.99 (0.99 to 0.99)** | 0.99 (0.99 to 0.99)* | 0.995 (0.991 to 0.999)* |
| Sex | | | |
| Female | Reference | Reference | – |
| Male | 0.60 (0.32 to 1.11) | 0.61 (0.33 to 1.15) | – |
| Living arrangement | | | |
| Cohabiting | Reference | Reference | Reference |
| Living alone | 0.40 (0.22 to 0.73)* | 0.43 (0.23 to 0.78)* | 0.54 (0.31 to 0.95) |
| Sex*living arrangement | 0.24 (0.07 to 0.81) | 0.24 (0.07 to 0.81) | – |
| Qualifications | – | 1.06 (0.78 to 1.45) | – |
| SIMD | – | 1.40 (1.05 to 1.87) | 1.47 (1.05 to 1.92)* |

Model 1 reports the ORs (95% CIs) for each psychological variable entered simultaneously in the same model, adjusting for age, age-squared, sex, living arrangement and sex*living arrangement. Model 2 reports the ORs (95% CIs) when additionally adjusting for deprivation and qualifications. Model 3 reports the results of a backwards elimination model using all variables entered in the fully adjusted model (model 2).
P values are false discovery rate corrected.
*P<0.05, **p<0.01, ***p<0.001.
Higher scores on SIMD reflect less deprivation.
Age$^2$, age-squared; SIMD, Scottish Index of Multiple Deprivation.

data (online supplemental table 4). Using only participants with complete data (n=384), the association between risk aversion and CRC screening was slightly stronger (OR=0.51, 95% CI 0.35 to 0.72; minimally adjusted) than that reported in table 2; however, the 95% CIs largely overlap. The effect sizes for the associations between the other psychological variables with CRC screening were similar in size to those reported in table 2. Despite similar effect sizes, the association between health literacy and intellect with CRC screening were no longer significant, possibly owing to the smaller sample size.

## DISCUSSION

This study investigated the association between risk aversion, present orientation, cognitive ability, health literacy and personality traits with CRC screening uptake and how these associations changed when all psychological variables were measured simultaneously. Using a sample of middle-aged and older adults living in Scotland, this study found that, when examining each psychological variable separately, participants who were more risk averse and more present oriented were less likely to have participated in CRC screening, whereas individuals who had higher health literacy, cognitive ability and intellect were more likely to have participated in CRC screening. When entering these psychological variables simultaneously in the same model, risk averse individuals were consistently less likely to have completed a home CRC screening test, even when additionally adjusting for socioeconomic indicators. Although cognitive ability was not associated with CRC screening in the fully adjusted model, a backwards elimination logistic regression retained cognitive ability as a correlate of CRC screening participation. Time preference, health literacy and intellect were no longer associated with CRC screening when all psychological characteristics were entered simultaneously, suggesting that these psychological variables do not have associations with CRC screening that are independent of the other psychological variables measured in the current study.

Finding that risk averse individuals were less likely to take part in CRC screening is in line with another study that reported that risk averse women were less likely to regularly undergo breast cancer screening.[23] Goldzahl[23] found that risk aversion was the most important predictor of breast cancer screening, accounting for 30% of disparity in screening regularity. The outcomes of treating cancer are uncertain. Risk averse individuals may be more sensitive to this uncertainty compared with risk seeking individuals.[23] Cancer screening can also lead to what might be perceived as risky treatments.[23 24] Emphasising how CRC screening can reduce risk of future ill-health, for example, by increasing survival rates and reducing the need for more invasive and uncertain cancer treatments, may encourage risk averse individuals to participate in CRC screening.[23]

The current study found some evidence that higher cognitive ability was associated with increased rates of screening. Completing a home CRC screening test may be a cognitively demanding task, and therefore individuals with lower cognitive ability might struggle to successfully complete this task. Gale *et al*[15] also found some support that higher cognitive ability predicted participation in a home CRC screening test. Similar to the current study, the association between cognitive ability and CRC screening was attenuated and non-significant when concurrently measuring other psychological variables (conscientiousness and health literacy) and indicators of socioeconomic status (household wealth).[15] These results suggest that the association between cognitive ability and CRC screening participation may not be independent of other psychological and socioeconomic variables.

Whereas other studies have found that being future oriented and more health literate are associated with an increased likelihood of participating in cancer screening,[18 22 23 38] the present study found that present orientation and health literacy associations were attenuated and non-significant when concurrently measuring other psychological variables. Present orientation and health literacy may not have associations with CRC screening that are independent of these other psychological characteristics. Conscientiousness, which has been linked to increased colorectal and other cancer screening in previous research,[14 15] did not predict CRC screening here. One limitation of the current study is that some of the measures had high levels of missing data. One-third of participants did not return the self-completion questionnaire, which included the personality assessment. Conscientiousness is characterised by being organised and industrious[35]; therefore, individuals who returned the questionnaire may score higher on conscientiousness than those who did not.

This study examined the association between a range of different psychological characteristics with CRC screening, but it did not consider the four key constructs (perceived susceptibility, severity, benefits and barriers) of the Health Belief Model. In addition to these constructs, the Health Belief Model assumes that modifying factors including psychological and socioeconomic variables can influence health behaviour.[10] These modifying factors may be mediators or moderators of the association between the Health Belief Model constructs and health behaviours.[10 25] Research should be carried out to explore the role that these psychological variables—especially risk aversion—play in the association between the Health Belief Model constructs and CRC screening participation.

This study has a number of advantages, including that the HAGIS pilot study was designed to be representative of Scottish adults aged over 50 years. However, only 42% of the analytic sample had data on all variables of interest. Participants with complete data on all variables were more likely to be cohabiting, have higher qualifications and live in less deprived areas; therefore, a limitation is that the sample used in the fully adjusted model is

not representative of the Scottish population. The wide range of psychological characteristics assessed in the HAGIS study is an advantage. This enabled the present study examine the role of each psychological characteristic on CRC screening when considered individually and concurrently with other psychological measures. A limitation is that this study examined whether participants have ever screened and not whether they regularly take part in CRC screening, which is important for detecting cancer early.[3] Future studies should investigate the psychological correlates of regularly participating in CRC screening.

In this sample of Scottish middle-aged and older adults, individuals who were risk averse were less likely to self-report having ever completed a home CRC screening test, even when adjusting for a range of other psychological and socioeconomic variables. Time preference, health literacy, cognitive ability and personality did not have associations with CRC screening participation independent of the other psychological characteristics measured. Educational materials that emphasise how CRC screening can reduce the risk of future ill-health may encourage risk averse individuals to participate in CRC screening.

**Contributors** CF-R contributed to the conception and design of the project, analysed the data, interpreted the data and drafted the initial manuscript. CBM contributed to the conception and design of the project, analysed the data, interpreted the data and drafted the initial manuscript. MvdP contributed to the statistical analysis of the data, interpreted the data and critically revised the manuscript. ED interpreted the data and critically revised the manuscript. DB interpreted the data and critically revised the manuscript. REO'C contributed to the conception and design of the project, interpreted the data and critically revised the manuscript. IJD contributed to the conception and design of the project, interpreted the data and critically revised the manuscript.

**Funding** This work was supported by the University of Edinburgh Centre for Cognitive Ageing and Cognitive Epidemiology, part of the cross council Lifelong Health and Well-being Initiative, funded by the Biotechnology and Biological Sciences Research Council and Medical Research Council (grant number: MR/K026992/1). The Healthy AGeing in Scotland (HAGIS) study (data resource) was part-funded by the National Institute of Ageing (grant number: RAG044535A) in the USA and was also part-funded by the Nuffield Foundation (grant number: OPD/42197). MvdP is supported by the Chief Scientist Office (CSO) of the Scottish Government Health and Social Care Directorates (grant number: not applicable).

**Disclaimer** The views expressed are those of the authors and not necessarily those of the foundation. The views expressed here are those of the unit and not necessarily those of the CSO.

**Competing interests** None declared.

**Patient and public involvement** Patients and/or the public were not involved in the design, or conduct, or reporting, or dissemination plans of this research.

**Patient consent for publication** Not applicable.

**Ethics approval** This study involves human participants and was approved by Ethical approval was obtained from the University of Stirling Ethics, the Administrative Data Research Network (PROJ-065-HAGIS Study), and the Public Benefits and Privacy Panel (HAGIS 1516-0417). Participants gave informed consent to participate in the study before taking part.

**Provenance and peer review** Not commissioned; externally peer reviewed.

**Data availability statement** Data are available on reasonable request. The anonymised HAGIS pilot data will be made available via the Gateway to Global Aging Data: https://g2aging.org/.

**ORCID iDs**
Chloe Fawns-Ritchie http://orcid.org/0000-0002-7493-2228
Elaine Douglas http://orcid.org/0000-0001-8540-1126

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
