## [Reviewer comments · BMJ Open]

ARTICLE DETAILS

TITLE (PROVISIONAL)	Psychological correlates of free colorectal cancer screening uptake in a Scottish sample: A cross-sectional observational study
AUTHORS	Fawns-Ritchie, Chloe; Miller, Christopher; van der Pol, Marjon; Douglas, Elaine; Bell, David; O'carroll, Ronan; Deary, Ian

VERSION 1 – REVIEW

REVIEWER	Anne Miles Birkbeck, University of London
REVIEW RETURNED	21-Sep-2020

GENERAL COMMENTS	This is a cross-sectional study examining factors associated with self-reported CRC screening uptake. Strengths of the paper include the following: the sample which aimed to be broadly representative of the population, and is a decent size to allow for the examination of numerous variables with sufficient power. The use of objective measures of cognitive function and health literacy. It is well written. The weaknesses are the limited literature review and the lack of reference to models of health behaviour. There are a number of studies that show associations between concepts from the Health Belief Model among others and screening uptake/interest such as perceived personal risk of cancer, perceived barriers and benefits of screening. The authors should reference this literature and might also like to read the paper by Whitaker et al (2011) on time perspective, barriers and screening. This limitation applies to both the introduction and discussion sections. It would be helpful if the authors presented a model of the associations they expected to see amongst their variables based on the existing literature, and for the authors to consider the interaction between variables such as risk aversion and future orientation – completing screening involves a risk of a bad outcome now while not completing screening involves the risk of a bad outcome later. It would also be useful to see whether the variables measured explain any of the associations between deprivation and CRC screening uptake, so perform a mediation analysis. Some of the decisions around the statistical analysis are unclear. So for example why age squared is entered and why interactions between sex and living arrangement. It's not clear why the authors use a backward elimination model – this kind of analysis is data rather than theory-driven. The authors need to provide more detail around missing data and why they have not performed any analyses to correct for missing data (imputation). Table 2 – it would be helpful if the reader could make sense of the table without reference to the text so where possible could the variables be labelled such that the direction of association is clear,
--

	for example you have put 'time preference' but it would be easier to follow if it were relabelled as either present or future –oriented.
--	--

REVIEWER	Amy Wahlquist East Tennessee State University
REVIEW RETURNED	22-Dec-2020

GENERAL COMMENTS	1. In the Abstract, the years of recruitment are listed as 2017-2018, but in Methods section under participants, it is listed as 2016-2017. Clarity here is needed to confirm when participants were recruited vs. participated in the HAGIS study as well as the current study. 2. The Abstract makes no mention of any demographics included in the analysis, though the analysis clearly included several demographic variables. Please mention this briefly in the Abstract. 3. In the Introduction, I think the word "screening" is left out in the second paragraph: "One study found that better cognitive function and higher conscientiousness, when measured in the same model, were associated with higher rates of participation in CRC [screening] after adjusting for age and sex. [10]". Please add 4. A CONSORT flow diagram would be helpful in following the flow of participants (sampled, recruited, enrolled/interviewed, self-completion questionnaire, missing data, etc). It is difficult to follow as these numbers appear in different sections and need to be referred to in order to follow where participants fell out of the flow. It is also important to include how many participants were sampled from the HAGIS study (2nd sentence in Participants section) or to clarify that the sampling was for the HAGIS study (reads like the first but not clear how many were sampled). 5. If socioeconomic status is known to be related to screening rates (Introduction), why is it not included as a demographic variable as age and sex were? Instead it was treated as an outcome? 6. The statistical analysis section was very detailed and explained the analyses performed in a thoughtful and appropriate way. Always enjoyable to read a manuscript where the methods are so clear. Thank you. Regarding the rank order correlations in Supplementary Table 1, it is not standard practice to evaluate a correlation for ordinal/binary variables (sex, cohabitating, qualifications). While I understand the need for simplicity in a table, the conclusions that would be drawn may not be the most appropriate. For example, when treating qualifications as a continuous measure, it is assuming that the difference between levels is the same, and that is probably not a fair assumption for this particular variable. It is recommended to remove these variables from the table (they are currently not mentioned in the results section) or include a different measure of agreement that is more appropriate for these types of variables. Seems strange to treat them as categorical/binary in Table 1 and then as pseudo-continuous in Supplementary Table 1. 7. Being part of the HAGIS study (since it was designed to be representative of middle-aged and older Scottish adults) is listed as an advantage, but not everyone who was included in HAGIS was included in this study. Even though HAGIS was designed to be representative, a sample from HAGIS would not necessarily have the same characteristics as being representative. Please acknowledge this potential limitation.
---

	8. Table 2 lists the OR for Risk Aversion as 0.66 but in the Results section of the text, the OR is listed as 0.67. Assuming a rounding issue - but please update for consistency.
--	--

VERSION 1 – AUTHOR RESPONSE

Reviewer 1

We thank Reviewer 1 for their comments on how to improve this manuscript. We have taken these comments on board and have amended the manuscript accordingly. We provide the response to each comment, below.

1. **This is a cross-sectional study examining factors associated with self-reported CRC screening uptake. Strengths of the paper include the following: the sample which aimed to be broadly representative of the population, and is a decent size to allow for the examination of numerous variables with sufficient power. The use of objective measures of cognitive function and health literacy. It is well written.**

The weaknesses are the limited literature review and the lack of reference to models of health behaviour. There are a number of studies that show associations between concepts from the Health Belief Model among others and screening uptake/interest such as perceived personal risk of cancer, perceived barriers and benefits of screening. The authors should reference this literature and might also like to read the paper by Whitaker et al (2011) on time perspective, barriers and screening. This limitation applies to both the introduction and discussion sections.

Response: In the introduction, we now describe the Health Belief Model and highlight some of the research examining the association between the HBM components (perceived susceptibility, severity, benefits and barriers) and cancer screening.

Pages 5-6: *“Psychological variables may also influence screening uptake. Many studies have used the Health Belief Model[6] to investigate whether perceptions about the susceptibility and severity of cancer and the perceived benefits and barriers to screening influence cancer screening uptake. The Health Belief Model purports that an individual will participate in a health behaviour, such as cancer screening, if they believe: 1) they are susceptible to the condition; 2) the condition is severe; 3) that there would be benefits to engaging in a certain health behaviour; and 4) that any benefits to the behaviour outweigh any barriers.[6-8] A study of 335 adults aged 50-79 years in the US, tested whether constructs of the Health Belief Model were associated with FOBT as well as other recommended CRC screening procedures in the US.[9] Specific barriers to completing the FOBT were associated with lower odds of having had a FOBT in the last 12 months; however, perceived CRC risk, perceived benefits of CRC screening in general, and perceived benefits of the FOBT were not associated with having had a FOBT in the last 12 months. The Health Belief Model has also been used to predict participation in other types of cancer screening. A review of studies using the Health Belief Model to predict uptake in mammography and pap screening found strong support that perceived benefits were positively associated with screening uptake and perceived barriers were negatively associated with screening uptake; however, they found weak support for an association between perceived susceptibility and severity with screening uptake.[10]*

Whereas a large number of studies have tested whether the constructs of the Health Belief Model are associated with cancer screening, the relationship between other psychological characteristics, such as cognitive ability and personality, with cancer screening is less well understood.”

We also now provide a description of the study by Whitaker et al. (2011) in the introduction:

Page 7: *“A study of 809 UK adults examined the role of time perspective in the association between socioeconomic inequalities and CRC screening uptake (flexible sigmoidoscopy).[25] This study did not find direct associations between high consideration of future consequences*

(future-orientation) and CRC screening uptake; however, the association between high socioeconomic status and CRC screening uptake was mediated by future-orientation. This study also found that future-orientation was positively associated with perceived benefits and negatively associated with perceived barriers of CRC screening, which were in turn associated with screening uptake.[25]

The discussion has been updated to make the reader aware that we did not consider the Health Belief Model in the current paper, and that future research could examine the relationship between the HBM constructs, the psychological characteristics assessed here (especially risk aversion), and CRC screening.

Page 20: *“This study examined the association between a range of different psychological characteristics with CRC screening, but it did not consider the four key constructs (perceived susceptibility, severity, benefits and barriers) of the Health Belief Model. In addition to these constructs, the Health Belief Model assumes that modifying factors including psychological and socioeconomic variables can influence health behaviour.[10] These modifying factors may be mediators or moderators of the association between the Health Belief Model constructs and health behaviours.[10, 25] Research should be carried out to explore the role that these psychological variables—especially risk aversion—play in the association between the Health Belief Model constructs and CRC screening participation.”*

Although we have updated the introduction and discussion to refer to the Health Belief Model, it is important to emphasise that this study was not designed to test health psychology models of health behaviour. It was designed to evaluate whether risk aversion, time preference, cognitive ability, health literacy and personality traits are associated with CRC screening, independent of the other psychological traits. We have now expanded the introduction to provide more information on the previous research undertaken on the associations between the psychological variables of interest (risk aversion, time preference, cognitive ability, health literacy and personality traits) and cancer screening participation to highlight to the reader the amount of research which has been conducted on this topic prior to the current analysis.

Page 6: *“Individuals with cognitive impairment are less likely to participate in CRC screening.[11, 12] There is some evidence that people who score higher on the personality traits extraversion and conscientiousness have higher rates of cancer screening.[11-15] The role of cognitive ability and personality in CRC screening were investigated together using a nationally representative sample of older adults from England.[15] Better cognitive ability and higher conscientiousness, when measured in the same model, were associated with higher rates of CRC screening participation after adjusting for age and sex. These associations were attenuated and non-significant when additionally adjusting for health literacy and household wealth.[15] Adequate health literacy—the ability to obtain and understand basic health information[16]—has been identified as another possible correlate of CRC screening. Whereas some studies have found that lower health literacy was associated with reduced rates of CRC screening, others have not.[17-20] Gale et al.[15] tested whether the association between health literacy and CRC screening may be partly explained by cognitive ability. The association between health literacy and CRC screening was attenuated by 40% when adjusting for cognitive ability.[15]”*

Page 7: *“Future-oriented women have been found to be more likely to undergo breast cancer and cervical screening.[22-24] However, when examining the role of time preference in prostate exam attendance, more present-oriented men were more likely to have had this procedure.[22] Characteristics associated with one screening programme may not necessarily be those associated with other types of screening [26].”*

Page 7-8: *“More risk averse individuals may be more likely to take part in screening because it can reduce the risk of ill-health by identifying cancer earlier, and therefore the treatment required is less aggressive and the prognosis is better. Alternatively, risk averse individuals may be less likely to take part in screening because they may see screening as risky.[23, 24] Positive screening results are likely to lead to medical treatments which have uncertain outcomes. Studies that have investigated the association between risk aversion and cancer screening provide some support for the latter hypothesis; that risk averse individuals are less likely to participate in cancer screening. More risk averse women have been found to be less likely to*

undergo regular breast cancer screening.[23] Another study[24] concluded that risk aversion was weakly associated with reduced use of cancer screening services; however, these associations did not reach statistical significance.”

- 2. It would be helpful if the authors presented a model of the associations they expected to see amongst their variables based on the existing literature, and for the authors to consider the interaction between variables such as risk aversion and future orientation – completing screening involves a risk of a bad outcome now while not completing screening involves the risk of a bad outcome later.**

Response: The aim of this study was to explore whether risk aversion, time preference, cognitive ability, health literacy and personality had associations with CRC screening that were independent of the other psychological traits. Previous studies have found associations between each of these variables and CRC screening participation. However, often only one of these variables, or a small number of them, are studied in relation to CRC screening. We also know that these psychological variables have moderate correlations with each other. Therefore, it is not clear from previous research which, if any, of these psychological characteristics would be associated with CRC screening when all were entered into the same model. We did not know which psychological variables would be associated with CRC screening when all psychological variables were included simultaneously and this is what we wanted to find out in the current study. We have updated the introduction to make it clear to the reader what the aim of the current study was.

Page 8: *“Most studies have tended to examine the association between only one psychological variable, or a small number of psychological variables, with cancer screening uptake. These psychological characteristics, however, have moderate correlations with each other. Higher cognitive ability is moderately correlated with health literacy[27] and with greater future-orientation.[28] Risk aversion correlates with lower cognitive function and lower literacy.[29] Risk aversion and time preference also have small, but inconsistent, associations with conscientiousness, agreeableness and extraversion.[30] It is not clear whether risk aversion, present-orientation, health literacy, cognitive ability, and personality traits, which are moderately correlated with each other, have associations with CRC screening uptake that are independent of the other psychological characteristics. Using data from the Healthy AGEing in Scotland (HAGIS) pilot study—a study designed to be representative of middle-aged and older Scottish adults—the present study aimed to test whether risk aversion, present-orientation, cognitive ability, health literacy and personality traits, when examined simultaneously in the same model, have independent associations with self-reported participation in home CRC screening.”*

We have now carried out an additional set of analyses to test whether there was an interaction between risk aversion and present-orientation, first adjusting for age, age-squared, sex, living arrangement, and sex*living arrangement, then additionally adjusting for deprivation and qualifications. This interaction was non-significant. We have updated the methods and results section to report this analysis:

Page 14: *“Finally, we tested whether there was an interaction between risk aversion and present-orientation. Completing a CRC screening test can involve the risk of a bad outcome in the short-term, whereas not completing a CRC screening test could involve the risk of a bad outcome in the future. Models were run which included risk aversion, present-orientation, and the interaction between risk aversion and present-orientation, first adjusting for age, age-squared, sex, living arrangement, and the interaction between sex and living arrangement, and then additionally adjusting for deprivation and qualifications.”*

Page 17: *“The interaction between risk aversion and present-orientation was non-significant (OR=1.04, 0.94 to 1.14; n=494) in a model including risk aversion, present-orientation, risk aversion*present-orientation, age, age-squared, sex, living arrangement, and sex*living arrangement. This interaction remained non-significant when additionally adjusting for deprivation and qualifications (OR=1.03, 0.94 to 1.13; n=490).”*

It would also be useful to see whether the variables measured explain any of the associations between deprivation and CRC screening uptake, so perform a mediation analysis.

Response: In the current analysis, deprivation was considered a covariate and not an exposure variable. The study sought to test whether psychological variables (exposure variables) were associated with CRC screening (outcome variables), independent of other psychological variables and independent of known demographic and socioeconomic variables (covariates). Testing whether the psychological variables explain any of the associations between deprivation and CRC screening uptake would therefore be asking a different question than the current study was designed to ask.

It may not have been clear that we were treating deprivation as a covariate from the way we had previously presented our results in Table 2 (i.e., we reported the ORs and 95% CI for the associations between deprivation and qualifications with CRC screening). We have therefore removed the deprivation and qualification rows from Table 2. In Table 2 (where psychological exposure variables are entered individually) we now report the ORs and 95% CIs for each exposure (risk aversion, time preference, cognitive ability, health literacy and personality traits) firstly adjusting for age, age-squared, sex, living arrangement, and sex*living arrangement, then additionally adjusting for deprivation and qualifications. This means we can examine the change in the size of the association between each of the psychological variables and CRC screening uptake before and after adjusting for indicators of SES. We think this is a more appropriate analysis to conduct to understand the relationship between psychological variables, SES and CRC screening in the context of our original aim. This also means that the order the covariates are entered in Table 2 is now the same as in Table 3 (when all psychological variables are entered simultaneously).

3. Some of the decisions around the statistical analysis are unclear. So for example why age squared is entered and why interactions between sex and living arrangement.

Response: An age-squared term was entered because the assumption of linearity of the logit was violated. An interaction term between sex and living arrangement was included because previous analysis (see below) using the HAGIS data found a significant interaction between sex and living arrangement with CRC screening uptake, such that co-habiting men have similar CRC screening rates as cohabiting and single women. Single men had lower rates of CRC screening uptake. The methods section has been updated to document why an age-squared term and an interaction between sex and living arrangement were added to the models:

Page 13: "An age-squared term was added as the assumption of linearity of the logit was violated. An interaction term between sex and living arrangement was included because previous research using the HAGIS data has found a significant interaction between sex and living arrangement on CRC screening uptake.[37] This study found that single men were less likely to take part in CRC screening, but there were no differences in CRC screening uptake for men living with a partner, single women or women living with a partner.[37]"

It's not clear why the authors use a backward elimination model – this kind of analysis is data rather than theory-driven.

Response: The main model in the current paper was the model in which all psychological variables were entered simultaneously to see which psychological variables had associations with CRC screening uptake that were independent of the other psychological variables. The backward elimination model was used to complement this main model. If the backward elimination model identified the same psychological variables as the model in which all psychological variables were entered simultaneously, this would provide additional support that this psychological variable was associated with CRC screening uptake. We have updated the methods section to make it clear to the reader why we have included a backward elimination logistic regression model:

Page 14: *“A backward elimination logistic regression was run to identify the best independent correlates of CRC screening participation. If the backward elimination model identified the same psychological variables as the models including all psychological variables entered simultaneously, this would provide additional support that this psychological characteristic was an important correlate of CRC screening participation.”*

The authors need to provide more detail around missing data and why they have not performed any analyses to correct for missing data (imputation).

Response: For each model conducted, we used participants with complete data on all variables included in the model. This meant there were some large differences in the number of observations entered into each of the models. The largest n was 867, whereas the smallest n was 384. Because of these differences in sample size across models, we re-ran all models presented in Table 2 using a subsample of participants with complete data on all variables of interest (n = 384). These models are reported in the supplementary materials (Supplementary Table 4). Generally, the effect sizes remained similar to those reported in the main text. The association between risk aversion and CRC screening, was slightly stronger using the smaller sample (OR=0.51, 95% CI 0.35 to 0.72; n = 384) than with the larger sample (OR=0.66, 0.51 to 0.85; n = 581), however these models have largely overlapping 95% CIs. We report the difference in the main text in the results section:

Page 18: *“Using only participants with complete data (n=384), the association between risk aversion and CRC screening was slightly stronger (OR=0.51, 0.35 to 0.72; minimally adjusted) than that reported in Table 2; however, the 95% CIs largely overlap. The effect sizes for the associations between the other psychological variables with CRC screening were similar in size to those reported in Table 2. Despite similar effect sizes, the association between health literacy and intellect with CRC screening were no longer significant, possibly owing to the smaller sample size.”*

We did not use imputation in the current study. One-third of participants did not return the self-completion questionnaire, meaning that we did not have data on a number of psychological variables of interest for these participants, including personality. Conscientiousness is characterised by being organised and industrious; therefore, it is likely that those who returned the self-completion questionnaire may score higher on conscientiousness than those who did not return it. The level of missing data, especially for the self-completion questionnaire, is a limitation in the current study. We have updated the methods section to state why we have not used imputation. We have also updated the discussion section to highlight that the level of missing data for some variables is a limitation of the current study:

Pages 14-15: *“Imputation was not used in the current study because those who did and did not return the self-completion questionnaire may differ on the psychological variables assessed in the self-completion questionnaire (e.g., conscientiousness).”*

Page 20: *“One limitation of the current study is that some of the measures had high levels of missing data. One-third of participants did not return the self-completion questionnaire, which included the personality assessment. Conscientiousness is characterised by being organised and industrious,[35] therefore individuals who returned the questionnaire may score higher on conscientiousness than those who did not.”*

Pages 20-21: *“However, only 42% of the analytic sample had data on all variables of interest. Participants with complete data on all variables were more likely to be cohabiting, have higher qualifications and live in less deprived areas, therefore a limitation is that the sample used in the fully-adjusted model is not representative of the Scottish population.”*

4. **Table 2 – it would be helpful if the reader could make sense of the table without reference to the text so where possible could the variables be labelled such that the direction of association is clear, for example you have put ‘time preference’ but it would be easier to follow if it were relabelled as either present or future –oriented.**

Response: We have changed “time preference” to “present-orientation” in Tables 2 and 3, and in the Supplementary tables.

Reviewer 2

We thank Reviewer 2 for their comments. We have made a number of changes to the manuscript following these comments. We think these changes have improved the manuscript. We provide a response to each comment, below.

1. **In the Abstract, the years of recruitment are listed as 2017-2018, but in Methods section under participants, it is listed as 2016-2017. Clarity here is needed to confirm when participants were recruited vs. participated in the HAGIS study as well as the current study.**

Response: HAGIS participants were recruited in 2016-2017. The 2017-2018 in the abstract was a typo and has now been updated to 2016-2017. Thank you for spotting this typo.

2. **The Abstract makes no mention of any demographics included in the analysis, though the analysis clearly included several demographic variables. Please mention this briefly in the Abstract.**

Response: The abstract has now been updated to make it clear that demographic information was included in the analysis.

Page 2: *“Controlling for age, age-squared, sex, living arrangement, and sex*living arrangement, a one-point increase in risk aversion (OR=0.66, 95% CI 0.51 to 0.85), and present-orientation (OR=0.86, 0.80 to 0.94) was associated with reduced odds of screening.”*

3. **In the Introduction, I think the word "screening" is left out in the second paragraph: "One study found that better cognitive function and higher conscientiousness, when measured in the same model, were associated with higher rates of participation in CRC [screening] after adjusting for age and sex. [10]". Please add**

Response: The word “screening” has been added so that the sentence now reads:

Page 6: *“Better cognitive ability and higher conscientiousness, when measured in the same model, were associated with higher rates of CRC screening participation after adjusting for age and sex.”*

4. **A CONSORT flow diagram would be helpful in following the flow of participants (sampled, recruited, enrolled/interviewed, self-completion questionnaire, missing data, etc). It is difficult to follow as these numbers appear in different sections and need to be referred to in order to follow where participants fell out of the flow. It is also important to include how many participants were sampled from the HAGIS study (2nd sentence in Participants section) or to clarify that the sampling was for the HAGIS study (reads like the first but not clear how many were sampled).**

Response: A flow diagram has now been added to the manuscript. We have also provided more information throughout the methods section to clarify how we arrived at 908 participants in the analytic sample, including making it clear that the sampling was for recruitment into the HAGIS study (and not deriving a random sample of HAGIS participants for the current analysis).

Pages 8-9: “Data from the HAGIS pilot study[31] were used here. To recruit participants, a random sample of 5,211 residential addresses in mainland Scotland were identified using the Postcode Address File. These addresses were screened by the National Records for Scotland to find addresses where at least one resident was aged over 50 years. A total of 3,088 addresses were identified with someone aged 50 years and older living in the household and these were the list of addresses fieldworkers used to recruit HAGIS participants.”

Page 9: “A total of 1,057 participants were recruited in 2016-2017 (Figure 1).”

Page 9: “Of the 1,057 participants who completed the face-to-face interview, 705 completed the self-completion questionnaire. Data collected during the face-to-face interview and the self-completion questionnaire were used in the current analysis.”

Page 9: “For the present report, the sample was restricted to participants aged between 50 and 80 years, as these participants would have been invited to take part in the Scottish Bowel Screening Programme. Of the 1,057 participants who completed the face-to-face interview, 917 participants were aged between 50 and 80 year.”

Page 13: “Of the 1,057 participants who completed the face-to-face interview, the following participants were removed before conducting analyses: participants who did not answer “yes” or “no” to the CRC screening question (no data, n=1; “don’t know”, n=6; “refused”, n=3); participants without data on age (n=3), participants aged <50 (n=7), participants aged >80 (n=126); participants without data on sex (n=1) or whose sex was recorded as transgender (n=1); participants without data on cohabiting status (n=1). We removed participants without data on CRC screening, age, sex, and living arrangement because these variables were entered into every model. After removal of these participants, the analytic sample consisted of 908 participants. A flow chart showing how the analytic sample was derived is shown in Figure 1.”

5. If socioeconomic status is known to be related to screening rates (Introduction), why is it not included as a demographic variable as age and sex were? Instead it was treated as an outcome?

Response: Indicators of socioeconomic status (i.e., deprivation and qualifications) were not treated as outcomes in the current study. In the current study, deprivation and qualifications were used as covariates like age and sex, rather than exposures like the psychological variables. It may not have been clear from the way we had previously presented the results in Table 2 that we were treating deprivation and qualifications as covariates and not exposures. In the previous version of this manuscript, Table 2 reported the ORs and 95% CIs for the association between deprivation and CRC screening, and between qualifications and CRC screening. We have now removed these rows from Table 2. We now firstly report the associations between each of the psychological variables with CRC screening in a minimally adjusted model (controlling for age, age-squared, sex, living arrangement and sex*living arrangement), and then a fully-adjusted model (additionally adjusting for deprivation and qualifications). By doing this, the order in which the covariates are entered into the models are the same when examining each psychological variable individually (Table 2) as those used when examining all psychological variables simultaneously (Table 3).

6. The statistical analysis section was very detailed and explained the analyses performed in a thoughtful and appropriate way. Always enjoyable to read a manuscript where the methods are so clear. Thank you. Regarding the rank order correlations in Supplementary Table 1, it is not standard practice to evaluate a correlation for ordinal/binary variables (sex, cohabitating, qualifications). While I understand the need for simplicity in a table, the conclusions that would be drawn may not be the most appropriate. For example, when treating qualifications as a continuous measure, it is assuming that the difference between levels is the same, and that is probably not a fair assumption for this particular variable. It is recommended to remove these variables from the table (they are currently not mentioned in the results section) or include a different measure of agreement that is more appropriate

for these types of variables. Seems strange to treat them as categorical/binary in Table 1 and then as pseudo-continuous in Supplementary Table 1.

Response: The primary purpose of the correlation matrix was to determine the strength of the correlation between the psychological variables, and these are the ones we have reported in the main text of the paper. However, it is common and good practice in epidemiological studies to include a correlation matrix between the predictor variables and covariables. It is useful to be able to compare the size of associations between predictors and covariates on a standard scale. We therefore have left the covariates in the correlation matrix reported in the supplementary materials. Correlations between continuous and binary variables are point-biserial correlations, and correlations between ordinal (ranked) variables and binary variables are rank-biserial correlations. We have updated the correlation matrix to include both Pearson and rank-order correlations (Supplementary Table 1). However, we note the difference in the size of the associations between the Pearson and rank-order correlations are very small, usually only changing the second decimal place.

7. Being part of the HAGIS study (since it was designed to be representative of middle-aged and older Scottish adults) is listed as an advantage, but not everyone who was included in HAGIS was included in this study. Even though HAGIS was designed to be representative, a sample from HAGIS would not necessarily have the same characteristics as being representative. Please acknowledge this potential limitation.

Response: A total of 1,057 participants took part in the HAGIS study. Participants who were not aged between 50 and 80 were removed because they would not have been invited to take part in CRC screening. After removal of these participants, the sample will still have been relatively representative of the Scottish population aged between 50 and 80 years; the population of interest in the current study. We also removed participants with missing data on the CRC screening question, age, sex, and living arrangement because these variables were included in all models. In total, only 15 participants were removed because of missing data on CRC screening, age, sex, and living arrangement. However, there was quite a lot of missing data for some of the variables, especially those assessed in the self-completion questionnaire. Only 384 participants had data on all variables. Participants with complete data were more likely to be cohabiting, have higher qualifications and live in less deprived areas compared to participants with missing data (reported in Supplementary Table 3). Therefore, the sample used in the fully-adjusted model will not be representative of the Scottish population.

Throughout the methods and results section, we have provided more information on the sample used here and the number of participants included at each stage (as described above). We have also updated the discussion to highlight that a limitation of the current study is that only 42% of the sample had data on all variables and therefore the sample used for the fully-adjusted model will not be representative of the Scottish population.

Pages 20-21: "This study has a number of advantages, including that the HAGIS pilot study was designed to be representative of Scottish adults aged over 50 years. However, only 42% of the analytic sample had data on all variables of interest. Participants with complete data on all variables were more likely to be cohabiting, have higher qualifications and live in less deprived areas, therefore a limitation is that the sample used in the fully-adjusted model is not representative of the Scottish population."

8. Table 2 lists the OR for Risk Aversion as 0.66 but in the Results section of the text, the OR is listed as 0.67. Assuming a rounding issue - but please update for consistency.

Response: The results section has been updated to OR=0.66 (Page 16).

VERSION 2 – REVIEW

REVIEWER	Amy Wahlquist East Tennessee State University
REVIEW RETURNED	13-May-2021

GENERAL COMMENTS	If eligible participants (aged >50) had partners over 45, were the partners allowed to be included in this study? Methods first paragraph implies so, but Figure 1 removes anyone under 50. And after removing those <50 (n=7) and >80 (n=126) and with no age (n=3), that would mean 921 (not 917 as mentioned) [were] interviewed aged between 50 and 80 years (last sentence of third paragraph in Methods to be update). Would also recommend adding the 705 that had self-completion questionnaires and 384 fully complete records to Figure 1.
--

VERSION 2 – AUTHOR RESPONSE

Reviewer 2

Reviewer comment: If eligible participants (aged >50) had partners over 45, were the partners allowed to be included in this study? Methods first paragraph implies so, but Figure 1 removes anyone under 50. And after removing those <50 (n=7) and >80 (n=126) and with no age (n=3), that would mean 921 (not 917 as mentioned) [were] interviewed aged between 50 and 80 years (last sentence of third paragraph in Methods to be update).

Author response: We thank reviewer two for highlighting that more clarity is needed in the manuscript when describing how we derived the sample used in the current analysis.

The eligibility criteria for taking part in the HAGIS study was that participants were to be aged over 50 years, or they were aged over 45 years and their partner had taken part in HAGIS. This meant that in the HAGIS sample there were a small number of participants aged under 50.

For the current paper, a subsample of HAGIS participants was used. We limited the age range to 50-80 years because participants outside of this age bracket would not have been invited to take part in bowel cancer screening.

Small edits to the methods and results section (and to Figure 1) have been made to make this clearer to the reader. We now make it explicit that the eligibility criteria outlined in the methods section refers to the eligibility criteria for taking part in HAGIS, we also now refer to the “HAGIS sample” in Figure 1.

Page 8: “Eligible participants for the HAGIS study were those aged over 50 years, and their partners if their partner was aged over 45 years.[31] A total of 1,057 participants took part in the HAGIS study in 2016-2017 and form the HAGIS sample (Figure 1)”

For clarity, we have moved reference to using a sub-sample of participants aged between 50 and 80 from the methods section and put this at the start of the results section when we are describing which HAGIS participants were excluded from the current analysis. This now means that the order in which we describe who was removed before carrying out the analysis in the main text matches that reported in Figure 1 and is the order in which we removed participants when carrying out the analysis.

Page 15: “Of the 1,057 participants who took part in HAGIS, the following participants were removed before conducting the current analyses: participants who did not answer “yes” or “no” to the CRC screening question (no data, n=1; “don’t know”, n=6; “refused”, n=3); participants without data on age (n=3), participants aged <50 (n=7), participants aged >80 (n=126); participants without data on sex (n=1) or whose sex was recorded as transgender (n=1); participants without data on cohabiting status (n=1). We removed participants without data on CRC screening, age, sex, and living arrangement because these variables were entered into every model. We removed participants aged under 50 and over 80 because these participants

would not have been invited to take part in the Scottish Bowel Cancer Screening Programme. After removal of these participants, the analytic sample consisted of 908 participants. A flow chart showing how the analytic sample was derived is shown in Figure 1.”

Reviewer comment: Would also recommend adding the 705 that had self-completion questionnaires and 384 fully complete records to Figure 1.

Author response: We have now updated Figure 1 to include those who form the sample with complete data (n = 384). We have also included in Figure 1 that 302 of the participants that make up the analytic sample did not complete the self-completion questionnaire and therefore these participants were removed before re-running the models with only participants with data on all variables of interest.

The n= 705 reported in the methods section refers to the number of the full HAGIS sample who completed the self-completion questionnaire. This is not the number of participants from the analytic sample used in the current study who returned the self-completion questionnaire. In total, 302 of the 908 participants who make up the analytic sample did not complete the self-completion questionnaire. We have updated the methods to make it clearer that the 705 refers to the number of participants in the entire HAIGS sample who returned the self-completion questionnaire.

Page 8: “The self-completion questionnaire covered topics including health behaviours and personality. Of the 1,057 participants who took part in HAGIS, 705 completed the self-completion questionnaire.”

We have also included the number of participants in the analytic sample who did not return the self-completion questionnaire.

Page 17: “Of the 908 participants who make up the analytic sample, 524 individuals had missing data on one or more of the predictor variables and covariates, including 302 participants who did not complete the self-completion questionnaire (Figure 1).”

These changes to the text, as well as to Figure 1, should clarify to the reader exactly how the HAGIS sample was derived as well as how the analytic sample used in the current study was derived.